# Intestinal Helminth Infection, Anemia, Undernutrition and Academic Performance among School Children in Northwestern Ethiopia

**DOI:** 10.3390/microorganisms10071353

**Published:** 2022-07-05

**Authors:** Abraham Degarege, Berhanu Erko, Yohannes Negash, Abebe Animut

**Affiliations:** 1Department of Epidemiology, College of Public Health, University of Nebraska Medical Center, Omaha, NE 68198, USA; 2Aklilu Lemma Institute of Pathobiology, Addis Ababa University, Addis Ababa P.O. Box 1176, Ethiopia; berhanu.erko@aau.edu.et (B.E.); yohannes.negash@aau.edu.et (Y.N.); abebe.animut@aau.edu.et (A.A.)

**Keywords:** helminth, hemoglobin level, anemia, undernutrition, academic performance

## Abstract

This study examined the prevalence and intensity of intestinal helminth infections and their association with anemia, undernutrition, and academic performance among school children in Maksegnit, northwestern Ethiopia. A total of 1205 school children, who attended Maksegnit Number Two Elementary School between May and July 2021, participated in this study. To determine helminth infection status, two thick Kato–Katz slides were examined for each child. Hemoglobin level was measured using a HemoCue machine. Academic performance was assessed using the mean score of all subjects children have taken for the Spring 2020/2021 academic term. Out of 1205 children examined, 45.4% were infected with at least one helminth species, 7.9% were anemic, and 35.8% were undernourished. The means for hemoglobin level and z-scores of weight for age, height for age, body mass index for age, and academic scores were lower among helminth-infected children than the uninfected. Children infected with intestinal helminths showed higher odds of anemia than those uninfected with helminths. In conclusion, there was a moderate prevalence of intestinal helminth infection and undernutrition among school children in Maksegnit. Intestinal helminth infection could increase the risk of anemia, undernutrition, and poor academic performance.

## 1. Introduction

Helminthiasis is a disease caused by parasitic worms that belong to cestodes (tapeworms), trematodes (flukes), nematodes (roundworms), and acanthocephalans (spiny-headed worms) [1,2,3]. These helminths have been infecting humans since prehistoric times and remain a persistent setback on the health, development, education, and economy of socially marginalized poor people in underdeveloped countries [1,2,3]. Persistence of helminthiasis in the poor could mainly be attributed to scarcity of basic health facilities along with limited access to clean water, soap, and latrine, which are the basis of hygiene. Handwashing with clean water and relevant soap can effectively prevent not only helminthic parasites’ ova/oocysts, but also several disease-causing protozoan, bacterial, viral, and fungal pathogens [1,2,3].

To date, the global prevalence of soil-transmitted helminth (STH) infection is estimated to be 1.5 billion, among which *Ascaris lumbricoides* constitutes to 807–1121 million, hookworms (*Ancylostoma duodenale* and *Necator americanus*) constitute 576–740 million, and *Trichuris trichiura* constitutes 604–795 million [4,5]. In 2019, over 236 million people required preventive treatment for schistosomiasis [6]. An estimated 95 to 135 million people in the world are also infected with *Taenia* spp. or *Hymenolepis nana* [7,8]. A recent meta-analysis study estimated a 33% prevalence of helminth infection among children in Ethiopia [9]. Recent studies in Ethiopia reported a high prevalence of intestinal helminth infection among school age children, reaching 70.8% in some regions [10,11]. 

Helminthiasis, in both endemic and traveler populations, could be mild and remain unrecognized in some or misdiagnosed in others in spite of its severe clinical outcomes. This makes the actual burden of helminthiasis and its impact significantly underestimated. Both asymptomatic and symptomatic cases develop chronic helminthiasis if not treated timely and effectively. Chronic helminthiasis causes protein–energy malnutrition, iron deficiency anemia, stunting of growth, cognitive impairment, organ damage, deficiency in vitamins (A, B6, B12) and minerals (iron, calcium, and magnesium), block nutrient absorption, and diminish immunity, thereby predisposing subjects to serious diseases [1,2,3]. *A. lumbricoides* causes intestinal obstruction, *Ascaris* pneumonitis, and hypersensitivity [12]. Hookworms cause chronic blood loss and depletion of iron store leading to iron deficiency anemia, which threatens the life of pregnant women and unborn children [13]. Trichuriasis causes ulceration of the intestine and iron deficiency anemia [13]. In addition, STH and schistosomiasis may cause learning, memory, and intelligence deficits in children [14,15]. 

Recognizing the severity of the burden of STHs and schistosomiasis and associated anemia, undernutrition, and poor academic performance among children in endemic regions, WHO’s member states proposed mass deworming to control infection with these parasites in a sustainable manner through existing health delivery channels. The strategy was endorsed by World Health Assembly in resolution WHA54.19 in May 2001. The resolution proposed regular treatment of 75–100% of all school-age children living in schistosomiasis, ascariasis, hookworm disease, and trichuriasis endemic areas by 2010 [16]. Ethiopia initiated its deworming program in 2007 by treating about 1 million school-aged children for schistosomiasis and STH and repeated the program in 2013 by treating 1.07 million school-age children. The Federal Ministry of Health of the country launched the national deworming program officially in November 2015 and treated approximately five million school-aged children against schistosomiasis in the same month. The five-year national program envisaged to control morbidity of schistosomiasis and STH by 2020 [17]. 

An update on the status of helminth infection and related morbidities, including anemia, undernutrition, and poor academic performance in children is needed to verify the effectiveness of the ongoing deworming program and make changes to the strategy if needed to ensure the WHO helminth control goals are being met. In addition, study findings on the relationship of helminth infection with undernutrition, anemia, and poor academic performance are mixed. While some studies reported increased risk of anemia [18,19,20,21], undernutrition [21,22,23,24], and poor academic performance [25,26,27] in children infected with helminths, several studies reported lack of relationship between infection with these parasites and anemia [28,29], undernutrition [30,31], and poor academic performance [32]. Even some studies showed lower odds of anemia and undernutrition among children infected with soil-transmitted helminths [33,34,35]. The aim of this study was to examine the prevalence and intensity of intestinal helminth infections and their association with anemia, undernutrition, and academic performance among school children in northwestern Ethiopia. 

## 2. Materials and Methods

### 2.1. Study Setting

The study was carried out in Maksegnit. Maksegnit is a small town located at about 40 kms from Gondar Town in the Amhara Regional State of Ethiopia. Children (age range: 6–16 years) attending Maksegnit Number Two Elementary School in Maksegnit Town, Gondar Zuria District between May and July 2021 participated in this study. Helminth infection is common in the town and the surrounding rural villages [36]. 

### 2.2. Study Design and Sampling Procedure 

As part of a longitudinal follow-up study aimed to evaluate the impact of repeated anthelminthic treatment infection on the epidemiology and clinical outcomes of malaria, we assessed the baseline prevalence of intestinal helminth infection and their association with anemia, undernutrition, and the academic performance of school children in Maksegnit. 

In order to recruit the study participants, the school vice director or supervisor and investigators entered to all sections of grades 1 to 7 in Maksegnit Number Two Elementary School, briefly explained the purpose of the study, and invited the students to participate in the study. Students who were willing to participate and whose parents provided consent provided finger prick blood and stool samples (about 5 mg), and they were measured their height and weight. 

### 2.3. Inclusion and Exclusion Criteria

Children eligible to participate in the study were those who (1) were aged 6–16 years; (2) were attending Maksegnit Number Two Elementary School from grade 1 to 7; (3) were residents of Maksegnit or the surrounding rural villages for four or more years; (4) have not taken any anthelmintic drug within six months before the start of the study; (5) have not received any antimalarial drug within one month before the beginning of the study; (6) volunteered to give stool samples and blood samples. On the other hand, children excluded from the study were those with a history of allergy to anthelmintic (praziquantel and albendazole) and antimalarial drugs (chloroquine or coartem).

### 2.4. Stool Sample Collection, Processing, and Diagnosis for Helminth Infection

A plastic sheet and applicator stick were given to the children to bring sizable stool (~5 mg). They were instructed to bring stool samples of their own in the labeled containers provided. Two Kato–Katz thick smears were prepared for each stool sample by the laboratory technicians following the WHO guideline [37]. The slides were examined to detect the presence in diagnostic stage of intestinal helminth infections within an hour of the Kato–Katz smear preparation in the field. Egg count for hookworms was simultaneously done on the spot within one hour of the smear preparation. The slides were then transported to Aklilu Lemma Institute of Pathobiology Medical Parasitology lab where they were examined quantitatively for eggs of other helminths species by an experienced laboratory technician. To estimate the egg counts per gram (epg) of stool, we multiplied the average egg counts of the two Kato–Katz slides by 24. The epg was used to determine the classes of intensity of infection following the WHO recommendation [38]. The WHO cut-off criterion below which indicates light intensity infection was 100 epg for *S. mansoni*; 1000 epg for *T. trichiura*; 2000 epg for hookworms; and 5000 epg for *A. lumbricoides*. The cut-off criterion below which indicates moderate intensity infection was 400 epg for *S. mansoni*; 10,000 epg for *T. trichiura*; 4000 epg for hookworms; and 50,000 epg for *A. lumbricoides*.

### 2.5. Nutritional Status

Age (in months), gender, height (to the nearest 0.1 cm), and weight (to the nearest 0.1 kg) measurements were entered into the WHO AnthroPlus software to calculate the Z scores for weight for age (WAZ), height for age (HAZ), and body mass index for age (BAZ) [39]. Using the Z values, children’s nutritional status was determined as underweight (WAZ < −2 and/or BAZ < −2), stunted (HAZ < −2), and undernourished (underweight and/or stunted).

### 2.6. Anemia Status 

Hemoglobin level, which was used to determine anemia status, was assessed from finger-prick blood using the HemoCue^®^ Hb 201+ System (https://www.hemocue.us/hb-201/. Accessed on 22 May 2022). Anemia was defined as a hemoglobin level <11 g/dL for children aged less than 12 years; <12 g/dL for children aged 12 to 14 years; <13 g/dL for male children aged 15 or 16 years [40].

### 2.7. Academic Performance 

Academic performance was assessed using the children’s mean score of all subjects they have taken in Spring 2020/2021 academic semester. The subjects from which the mean scores calculated were Amharic, English, Maths, Environmental Science, and Arts for students who attended grade 1 to 4; Amharic, English, Maths, Science, Social Science, Ethics, Music, Art, and Sport for those attending grade 5 and 6; and Amharic, English, Maths, Social Science, Ethics, Sport, Biology, Physics, and Chemistry for students attending grade 7. The students’ academic performance in each subject was graded on a scale of 100%. 

### 2.8. Data Analysis 

Stata (V 16) was used to analyze the data. A binary association of helminth infection and demography factors was conducted using a chi-square test. Correlation of intestinal helminth infection with the hemoglobin level, WAZ, HAZ, BAZ, and academic score was examined using a linear regression analysis. A logistic regression model was used to compare the odds ratio of anemia, stunting, underweight, and undernutrition among children infected vs. uninfected with intestinal helminths after controlling for demography status. Odds ratio and mean differences were considered significant when *p*-values were <0.05.

### 2.9. Ethical Approval and Consent Process 

The protocol for this study was approved by the institutional review boards of University of Nebraska Medical Center (IRB # 0618-20-FB) and Aklilu Lemma Institute of Pathobiology at Addis Ababa University (ALIPB IRB/25/2012/20). Permission to conduct the study was also obtained from the Amhara regional health office, Central Gondar Zonal Health office, and Gondar Zuria district health office. Permission was also obtained from the Central Gondar Zonal Educational office and the director of the Maksegnit Number Two Primary School. Children were asked to bring their parents or guardians to the school. Purpose of the study, procedures to be followed, and potential side effects were explained to parents/guardians and asked if they were willing to have their child(ren) be part of the study. Written informed consent from willing parents and assent from children were obtained prior to the start of the investigation.

## 3. Results

### 3.1. Prevalence of Intestinal Helminth Infection

A total of 2172 children attending grade 1 to grade 7 at Maksegnit Number Two Elementary School were invited to participate in the study, of which 1246 agreed and provided demographic, height, and weight data. Some children could not provide stool (*n* = 41) or blood sample (*n* = 58). A total of 1205 children brought stool sample, of which 29.5% were infected with *Schistosoma mansoni*, 22.2% with *A. lumbricoides*, 7.3% with hookworms, 4.2% with *H. nana*, 1.7% with *Enterobius vermicularis*, 1.6% with *Taenia* spp., 0.17% with *T. trichiura*, and 45.4% with at least one of these parasite species (Table 1). The prevalence of infection with at least one helminth species was significantly (*p* < 0.01 for both) greater among males (50.8%) than females (41.0%) and among children aged 11 to 16 years (50.9%) than in those of 6 to 10 years old (39.75%). *S. mansoni* infection was significantly (*p* < 0.01) greater among males (35.8%) than females (24.3%) and among children of 11 to 16 years (33.2%) than those of 6 to 10 years old (25.6%). The prevalence of *A. lumbricoides* infection was greater among children in the 11 to 16 years age group (25.4%) than those in 6 to 10 years age group (18.9%). Infection with at least one helminth species was most prevalent in children attending grade 6 (55.6%) and least common in those attending grade 1 (40.2%) (*p* = 0.038). The prevalence of *S. mansoni*, hookworms, and *H. nana* infections also showed significant variation across the children’s grades. The mean egg per gram of stool for the most prevalent helminths seen among the children were 3003.5 (range = 24, 21,600) for *A. lumbricoides*, 264.30 (range = 24, 3768) for *S. mansoni*, and 2618.8 (range = 24, 23,904) for hookworms. The majority of *A. lumbricoides* (80.0%), hookworms (64.8%), and *S. mansoni* (45.6%) cases had light intensity infections. Close to 15% of children infected with hookworms and *S. mansoni* had heavy intensity infections. None of the children were heavily infected with *A. lumbricoides*.

### 3.2. Intestinal Helminth Infection and Hemoglobin Level

Mean hemoglobin level of children infected with helminths ranged from 12.4 g/dL (infected with four species) to 12.8 g/dL (infected with *A. lumbricoides*) (Table 2). The mean hemoglobin level was significantly lower among children infected with at least one intestinal helminth species (12.75 g/dL) compared to those who were not infected with helminth (12.93 g/dL) (adjusted mean difference (β) = −0.16, 95% CI = −0.30, −0.02). Individuals infected with only one intestinal helminth species (β = −0.16, 95% CI = −0.32, −0.01) particularly with *S. mansoni* (β = −0.15, 95% CI = −0.30, 0.00) had lower hemoglobin level than those free from helminth infection. Individuals infected with only *A. lumbricoides*, hookworms, *H. nana*, *E. vermicularis*, or *Taenia* spp. and those infected with two or more of these species also showed lower hemoglobin level than the uninfected ones, although differences were not significant. The mean hemoglobin level was significantly lower among males than females (β = −0.27, 95% CI = −0.40, −0.13) and among those aged 6 to 10 years than those in the 11 to 16 years age group (β = −0.33, 95% CI = −0.46, −0.19) after adjusting for helminth infection and nutritional status. 

### 3.3. Intestinal Helminth Infection and Anemia

Out of 532 individuals infected with helminth, 11.3% were anemic, but only 5.1% of the 649 children who were not infected with helminths were anemic. The highest prevalence of anemia was seen among children infected with hookworms (13.8%) (Table 3). The odds of anemia were greater among children infected with at least one intestinal helminth species than those uninfected with helminths (adjusted odds ratio, aOR = 1.90, 95% CI = 1.20, 2.99). Children who were infected with only one intestinal helminth species (aOR = 1.80, 95% CI = 1.08, 2.99) particularly with hookworms (aOR = 1.99, 95% CI = 1.01, 3.92), *E. vermicularis* (aOR = 3.46, 95% CI = 1.08, 11.11), and *Taenia* spp. (aOR = 3.93, 95% CI = 1.31, 11.83) showed higher odds of anemia compared to those who were uninfected with helminths. The odds of anemia were also greater among children infected with two or more helminth species than those uninfected with helminths (aOR = 2.10, 95% CI = 1.19, 3.71). The odds of anemia among children infected with *S. mansoni* increased with an increase in the epg of the parasite (aOR = 1.001, 95% CI = 1.00002, 1.001). The odds of anemia were greater among children who were males vs. females (aOR = 3.29, 95% CI = 2.04, 5.29), with ages 11 to 16 years vs. 6 to 10 years (aOR = 1.71, 95% CI = 1.09, 2.69), and undernourished ones vs. normal (aOR = 1.57, 95% CI = 1.01, 2.44).

### 3.4. Helminth Infection and Nutritional Status

Among the 532 children infected with helminth, 15.5% were stunted (HAZ < −2), 26.8% were underweight (WAZ < −2 and/or BAZ < −2), and 35.8% were undernourished (stunted and/or underweight) (Table 4). The corresponding values among individuals who were not infected (*n* = 649) were 12.1%, 26.9%, and 33.2%, respectively. The differences in the prevalence of stunting, underweight, and undernutrition by the species and number of helminth infections was not statistically significant. However, the odds of stunting (aOR = 2.87, 95% CI = 1.34, 6.18) and undernutrition (aOR = 2.28, 95% CI = 1.19, 4.40) among individuals infected with *S. mansoni* were greater in those with heavy than light intensity infection. The odds of underweight (aOR = 3.77, 95% CI = 1.05, 13.48) and undernutrition (aOR = 3.36, 95% CI = 1.05, 10.77) among individuals infected with hookworms were also greater in those with moderate than light intensity infection. The mean WAZ (β = −0.20, 95% CI = −0.36, −0.05), HAZ (β = −0.15, 95% CI = −0.26, −0.04), and BAZ (β = −0.27, 95% CI = −0.40, −0.13) were significantly lower among infected children than the uninfected with helminths (Table 5). The difference in the mean WAZ (β = −0.19, 95% CI = −0.37, −0.02) and HAZ (β = −0.17, 95% CI = −0.29, −0.05) between individuals infected vs. uninfected with helminth was particularly significant among individuals infected with *S. mansoni* vs. those uninfected with helminth. Children who were co-infected with two or more helminth species showed significantly lower HAZ (β = −0.22, 95% CI = −0.38, −0.05) than those uninfected with helminths.

### 3.5. Helminth Infection and Children’s Academic Score

Children infected with at least one intestinal helminth species had a mean academic score of 66.4% from the expected maximum score of 100%, while those not infected had a mean academic score of 69.2% (Table 6). The mean score difference between the infected and uninfected was significant after controlling for age, gender, and nutritional status (β = −2.22, 95% CI = −3.69, −0.75). The difference in the mean academic score was particularly greater in those co-infected with two or more helminth species at a time (β = −3.82, 95% CI = −5.85, −1.79). The mean academic score of children infected with at least one intestinal helminth species ranged from 58.7% in those infected with four species to 73.7% in those infected with *Taenia* spp. Furthermore, children infected with *S. mansoni* (β = −2.97, 95% CI = −4.61, −1.33) or *A. lumbricoides* (β = −2.21, 95% CI = −4.20, −0.22) showed lower mean academic score compared to those without helminth infection. The mean score was greater among males than females (β = 2.28, 95% CI = 0.82, 3.74), and among those 6 to 10 years old than those aged 11 to 16 years (β = −1.14, 95% CI = −2.60, 0.32).

## 4. Discussion

Intestinal helminth infections were prevalent and associated with anemia, undernutrition, and low academic performance in children who attended Maksegnit Number Two Primary School in northwestern Ethiopia. About 45% of the 1205 study participants were infected with at least one intestinal parasite, 15% of the infected were anemic, and 35.8% were undernourished. Intestinal helminth infection was associated with decreased mean hemoglobin level, increased odds of anemia, decreased WAZ, HAZ, and BAZ, and low academic score. Correlation of *S. mansoni* infection with low hemoglobin, WAZ, HAZ, and academic score was strong. In addition, anemia was strongly associated with hookworms, *E. vermicularis*, and *Taenia* spp. infections.

The prevalence of intestinal helminth infection in the current study population was lower compared to the estimates we reported in the same population age group in different regions of Ethiopia (Range:54.9 to 58.3%) [41,42]. A recent meta-analysis estimated a 33% pooled prevalence of soil-transmitted helminth infection among school children [9]. This corroborates the high prevalence of helminth infection among school-age children in the country despite the ongoing regular deworming programs to control transmission of the parasites effectively [43]. This could result from limited antihelminth drugs and funds to support the deworming program in accordance with the WHO guideline and hence reinfection of children shortly after treatment. 

Helminth infection was associated with low hemoglobin level and increased risk of anemia. The association was significant in the case of *S. mansoni*, hookworms, *E. vermicularis*, and *Taenia* spp. infections. Previous reports also documented association of hookworms [18,19,24] and *S. mansoni* infections [19] with low hemoglobin level or increased risk of anemia. Altogether, these findings support that helminth infections impact blood hemoglobin levels differently, leading to iron deficiency anemia. For instance, the hookworm sucks blood, releases anticoagulants, causes bleeding in the intestinal wall, and competes with the host for nutrition [13]. *S. mansoni* infection causes autoimmune hemolysis, extra-corporeal loss of iron, and splenic sequestration, leading to iron deficiency anemia [44]. The mechanisms by which *E. vermicularis* and *Taenia* spp. infections affect hemoglobin level are unclear apart from inducing low appetite and hence reduced food uptake and malabsorption in the intestine, leading to iron deficiency anemia. 

Intestinal helminth infection was associated with lower WAZ, HAZ, and BAZ and was magnificent in the case of *S. mansoni* infection. Intestinal helminth infections can directly or indirectly affect nutritional status, leading to undernutrition [24,45,46]. Intestinal helminths can trigger gastrointestinal tract physiopathology and low appetite, decreasing food and iron uptake [47,48]. Intestinal helminths also secrete potent inhibitors of pancreatic enzymes that block host nutrient absorption in the small intestine [49]. Helminths can also induce intestinal inflammation [50] and chronic blood loss [51], leading to undernutrition. However, the difference in WAZ, HAZ, and BAZ between infected vs. uninfected was statistically insignificant upon analyzing nutritional status as a categorical variable (i.e., underweight/stunted vs. normal). It could be possible that the small differences captured when z values were treated as a continuous variable could be lost when merged into two groups. 

In this study, intestinal helminth infection was correlated with low academic performance. Studies in South Africa, Ghana and Egypt also showed lower academic performance in children infected with soil transmitted helminths compared to their non infected counterparts [25,26,52]. A meta-analysis of 15 studies also reported lower scholastic achievement/academic scores among children infected with *Schistosoma* and/or those not dewormed with praziquantel compared to uninfected ones [15]. Helminth infection could negatively affect school attendance, leading to low academic performance [15,53]. *Schistosoma* infection could deposit eggs in the central nervous system and cause physical discomfort/distraction, anemia, and undernutrition that potentially affect cognitive performance, leading to low academic score [54,55].

One of the strengths of this study was the inclusion of a large enough sample size that provided sufficient power to test the relationship of helminth infection with hemoglobin level, nutritional status, and academic performance. In addition, the majority of the participants contacted participated in this study, increasing the generalizability of the findings. However, we did not control the nutritional uptake of children and their parent’s socioeconomic status, including income and education status, which could potentially affect the children’s hemoglobin level and nutritional status. In addition, factors such as study habits and the nature of the school’s examinations that may affect students’ scores were not controlled while examining the relationship between helminth infection and academic scores. Furthermore, although standard method (two Kato–Katz slides) was used for the diagnosis of *S. mansoni* infection, supplementing the test results with other techniques such as circulating cathodic antigen (CCA) could have improved the accuracy of the test results.

## 5. Conclusions

In conclusion, there was a moderate prevalence of intestinal helminth infection among school children in Maksegnit and its surrounding rural villages in northwestern Ethiopia. This could increase the risk of anemia, undernutrition, and poor academic performance in the area and beyond. Thus, there is a need for integrated helminthiasis control program aimed at reducing transmission of infection and related morbidities in order to improve children’s health and their academic performance. 

## Figures and Tables

**Table 1 microorganisms-10-01353-t001:** Percent prevalence of intestinal helminth infection among school children in Maksegnit in May to July 2021 (N = 1205).

	Number Examined	*S. mansoni*	*A. lumbricoides*	Hook Worms	*T. trichiura*	*H. nana*	*Taenia* spp.	*E. vermicularis*	At Least One Helminth Species
**Age group in years**									
6–10	600	25.6	18.9	6.4	0.2	4.0	1.3	2.0	39.7
11–16	605	33.2	25.4	8.4	0.2	4.5	1.8	1.5	50.9
*p*-value		**0.004**	**0.007**	0.189	0.998	0.677	0.492	0.504	**<0.001**
**Gender**									
Females	666	24.3	20.4	7.2	0.2	3.5	1.2	1.7	41.0
Males	539	35.8	24.5	7.4	0.2	5.2	2.0	1.9	50.8
*p*-value		**<0.001**	0.091	0.887	0.881	0.135	0.245	0.788	**0.001**
**Grade**									
1	174	19.5	20.7	4.0	0.6	8.1	1.7	2.3	40.2
2	233	29.6	18.0	6.9	0.0	3.9	1.3	2.6	42.1
3	215	34.4	24.2	8.8	0.0	2.3	0.9	1.4	48.8
4	202	31.7	21.3	8.4	0.0	1.5	2.0	0.0	40.6
5	170	32.9	22.4	4.1	0.0	5.9	0.6	1.8	45.9
6	178	28.7	27.5	11.8	0.6	5.6	3.4	2.8	55.6
7	33	21.2	24.2	3.0	0.0	0.0	0.0	0.0	45.5
*p*-value		**0.037**	0.405	**0.048**	0.562	**0.017**	0.398	0.344	**0.038**

Bold values indicate that the differences were statistically significant.

**Table 2 microorganisms-10-01353-t002:** Mean hemoglobin level difference by helminth infection status, nutritional status, age group, and gender among school children in Maksegnit from May to July 2021 (N = 1188).

Variables	Frequency	Mean Hemoglobin Level	Adjusted Mean Hemoglobin Level Difference (95% CI)
**Helminth infection status**			
Uninfected with helminth	649	12.93	Ref
*S. mansoni*	343	12.72	**−0.15 (−0.30, 0.00)**
*A. lumbricoides*	261	12.81	−0.05 (−0.21, 0.11)
Hookworms	87	12.80	−0.10 (−0.36, 0.15)
*H. nana*	50	12.68	−0.14 (−0.47, 0.20)
*E. vermicularis*	21	12.50	−0.30 (−0.80, 0.21)
*Taenia* spp.	19	12.51	−0.35 (−0.88, 0.18)
Multiple helminths			
Only one helminth species	334	12.75	**−0.16 (−0.32, −0.01)**
Any two helminth species	148	12.78	−0.12 (−0.33, 0.09)
Any three helminth species	47	12.68	−0.25 (−0.60, 0.10)
Any four helminth species	3	12.40	−0.43 (−1.76, 0.91)
Any helminth	532	12.75	**−0.16 (−0.30, −0.02)**
**Gender**			
Females	657	12.98	
Males	531	12.68	**−0.27 (−0.40, −0.13)**
**Age group**			
11 to 16 years	596	13.01	
5 to 10 years	592	12.70	**−0.33 (−0.46, −0.19)**
**Nutritional status**			
Normal	784	12.90	
Undernourished	404	12.78	−0.13 (−0.27, 0.01)

Bold values indicate that the differences were statistically significant.

**Table 3 microorganisms-10-01353-t003:** Comparison of the odds of anemia by helminth infection status, nutritional status, age group, and gender among school children in Maksegnit from June to July 2021 (N = 1188).

Variables	Frequency	Prevalence of Anemia (%)	Adjusted Odds of Anemia (95% CI)
**Helminth infection status**			
Uninfected with helminth	649	5.08	Ref
*S. mansoni*	343	10.79	1.24 (0.78, 1.95)
*A. lumbricoides*	261	11.49	1.55 (0.96, 2.51)
Hookworms	87	13.79	**1.99 (1.01, 3.92)**
*H. nana*	50	8.00	0.86 (0.29, 2.48)
*E. vermicularis*	21	19.05	**3.46 (1.08, 11.11)**
*Taenia* spp.	19	26.32	**3.93 (1.31, 11.83)**
Multiple helminths			
Only one helminth species	334	10.48	**1.80 (1.08, 2.99)**
Any two helminth species	148	12.84	**2.13 (1.14, 3.96)**
Any three helminth species	47	10.64	1.70 (0.61, 4.72)
Any four helminth species	3	33.30	4.68 (0.40, 54.69)
Any helminth	532	11.28	**1.90 (1.20, 2.99)**
**Gender**			
Females	657	3.96	**----**
Males	531	12.81	**3.29 (2.04, 5.29)**
**Age group**			
6 to 10 years	592	5.78	
11 to 16 years	596	9.97	**1.71 (1.09, 2.69)**
**Nutritional status**			
Normal	784	6.57	**----**
Undernourished	404	10.40	**1.57 (1.01, 2.44)**

Bold values indicate that the differences were statistically significant.

**Table 4 microorganisms-10-01353-t004:** Helminth infection status and nutritional status among school children in Maksegnit from May to July 2021 (N = 1205).

Helminth infectionstatus	Frequency	Prevalence of Stunting (%)	Adjusted Odds Stunting (95% CI)	Prevalence of Underweight (%)	Adjusted Odds Underweight (95% CI)	Prevalence of Undernourished (%)	Adjusted Odds Undernourished (95% CI)
Uninfected with helminth	649	12.08	Ref	26.91	Ref	33.18	Ref
*S. mansoni*	343	16.19	1.35 (0.95, 1.91)	10.79	1.12 (0.84, 1.49)	10.79	1.22 (0.94, 1.57)
*A. lumbricoides*	261	14.34	1.08 (0.73, 1.60)	11.49	1.00 (0.70, 1.42)	11.49	1.02 (0.77, 1.36)
*T. trichiura*		0.0	----	100.0			
Hookworms	87	14.8	1.11 (0.60, 2.04)	13.79	0.84 (0.48, 1.47)	13.79	0.93 (0.59, 1.48)
*H. nana*	50	15.7	1.19 (0.55, 2.57)	25.49	0.91 (0.48, 1.73)	8.00	1.04 (0.58, 1.87)
*E. vermicularis*	21	4.8	0.31 (0.04, 2.34)	14.29	0.45 (0.13, 1.55)	19.05	0.31 (0.09, 1.07)
*Taenia* spp.	19	5.3	0.35 (0.05, 2.63)	26.32	0.79 (0.26, 2.44)	26.32	0.88 (0.33, 2.33)
Multiple helminths							
Only one helminth	334	16.52	1.44 (0.99, 2.09)	27.7	1.04 (0.78, 1.40)	37.5	1.21 (0.92, 1.59)
Any two helminth	148	15.03	1.29 (0.78, 2.13)	21.6	0.75 (0.49, 1.14)	30.7	0.89 (0.61, 1.31)
Any three helminth	47	8.51	0.68 (0.24, 1.94)	38.3	1.69 (0.91, 3.11)	40.4	1.37 (0.75, 2.50)
Any four helminth	3	33.3	3.64 (0.33, 40.60)	0.0	----	33.3	1.01 (0.09, 11.17)
Any helminth	532	15.50	1.33 (0.96, 1.86)	26.8	0.99 (0.77, 1.28)	35.8	1.07 (0.84, 1.37)

**Table 5 microorganisms-10-01353-t005:** Correlation of helminth infection status with HAZ, WAZ, and BAZ among school children in Maksegnit from May to July 2021 (N = 1205).

Helminth Infection Status	Frequency	Mean HAZ	Adjusted Mean HAZ Difference (95% CI)	Mean WAZ	Adjusted Mean WAZ Difference (95% CI)	Mean BAZ	Adjusted Mean BAZ Difference (95% CI)
Uninfected with helminth	649	−0.89	Ref	−1.16	Ref	−1.25	Ref
*S. mansoni*	343	−1.15	**−0.17 (−0.29, −0.05)**	−1.40	**−0.19 (−0.37, −0.02)**	−1.34	−0.07 (−0.20, 0.06)
*A. lumbricoides*	261	−1.03	−0.01 (−0.14, 0.12)	−1.37	−0.15 (−0.35, 0.04)	−1.31	−0.03 (−0.17, 0.11)
*T. trichiura*		−1.71	−0.74 (−2.08, 0.59)	−2.87	−1.61 (−3.47, 0.25)	−2.25	−0.97 (−2.39, 0.44)
Hookworms	87	−1.03	−0.004 (−0.21, 0.21)	−1.45	0.20 (−0.51, 0.11)	−1.41	−0.14 (−0.36, 0.08)
*H. nana*	50	−1.21	−0.23 (−0.50, 0.04)	−1.47	−0.26 (−0.65, 0.12)	−1.31	−0.03 (−0.31, 0.26)
*E. vermicularis*	21	−0.97	−0.08 (−0.49, 0.34)	−1.14	0.45 (−0.47, 0.62)	−1.17	0.09 (−0.35, 0.53)
*Taenia* spp.	19	−1.07	−0.01 (−0.44, 0.43)	−0.83	0.44 (−0.22, 1.10)	−1.12	0.19 (−0.27, 0.65)
Multiple helminths							
Only one helminth	334	−1.12	**−0.17 (−0.30, −0.04)**	−1.37	**−0.21 (−0.39, −0.03)**	−1.30	−0.03 (−0.17, 0.10)
Any two helminth	148	−1.05	−0.10 (−0.27, 0.07)	−1.26	−0.09 (−0.35, 0.16)	−1.20	0.07 (−0.11, 0.25)
Any three helminth	47	−1.05	−0.08 (−0.37, 0.20)	−1.62	**−0.43 (−0.83, −0.03)**	−1.69	**−0.41 (−0.71, −0.11)**
Any four helminth	3	−2.02	−0.95 (−2.04, 0.14)	−1.51	−0.27 (−2.13, 1.58)	−1.20	0.11 (−1.05, 1.26)
Any helminth	532	−1.10	**−0.15 (−0.26, −0.04)**	−1.37	**−0.20 (−0.36, −0.05)**	−1.37	−0.03 (−0.15, 0.08)

Bold values indicate that the differences were statistically significant.

**Table 6 microorganisms-10-01353-t006:** Helminth infection status and academic score among school children in Maksegnit from May to July 2021 (N = 1205).

Variables	Mean Average Score	Adjusted Mean Average Score Difference (95% CI)
**Helminth infection status**		
Uninfected with helminth	69.6	Ref
*S. mansoni*	66.3	**−2.97 (−4.61, −1.33)**
*A. lumbricoides*	66.4	**−2.21 (−4.20, −0.22)**
Hookworms	67.5	0.92 (−2.16, 4.00)
*T. trichiura*	58.1	−11.97 (−29.92, 5.96)
*H. nana*	69.7	1.27 (−2.29, 4.84)
*E. vermicularis*	66.0	−2.63 (−8.13, 2.86)
*Taenia* spp.	73.7	5.82 (−0.01, 11.66)
Only one helminth species	68.1	−1.33 (−3.01, 0.35)
Any two helminth species	66.3	**−3.42 (−5.69, −1.15)**
Any three helminth species	65.5	**−4.24 (−8.03, −0.47)**
Any four helminth species	58.7	−11.25 (−25.71, 3.21)
Any helminth	12.75	**−2.22 (−3.69, −0.75)**
**Gender**		
Females	67.6	
Males	69.8	**2.28 (0.82, 3.74)**
**Age group**		
6 to 10 years	69.5	
11 to 16 years	67.8	**−1.14 (−2.60, 0.32)**
**Nutritional status**		
Normal	69.1	
Undernourished	67.8	−0.95 (−2.48, 0.58)

Bold values indicate that the differences were statistically significant.

## Data Availability

The data presented in this study are available on request from the corresponding author. The data are not publicly available due to privacy/ethical issues.

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
