# Peer review of "Intestinal Helminth Infection, Anemia, Undernutrition and Academic Performance among School Children in Northwestern Ethiopia"

_microorganisms, 2022, doi:10.3390/microorganisms10071353_

Round 1

Reviewer 1 Report

The paper is very interesting and important for public health. It prevents prevalence results and their association with the parameters examined, but it should also include results on the mean intensity and range of intensity of infection or on relative abundance (see Bush et al. 1997). Do the authors have data on the intensity of infection for each parasite species in each child?

Page 2, Lines 83-86: This should be rewritten to express the aim of the study. What was the aim of the study?

Tables 1, 2, 3 – ‘Number’ should be clarified: number of what?

Page 10, Lines 321-324: This sentence explains why the link between the intensity of infection and the parameters examined was not analyzed, but examination of intensity should be included in the “aim of the study”, and the results for the intensity of infection of individual parasite species should be given in the “Results” section. The information that the intensity of infection was low or moderate, given in the “Discussion” section, is insufficient.

Author Response

Reviewer 1

  • The paper is very interesting and important for public health. It prevents prevalence results and their association with the parameters examined, but it should also include results on the mean intensity and range of intensity of infection or on relative abundance (see Bush et al. 1997). Do the authors have data on the intensity of infection for each parasite species in each child?

Response: Thank you for the useful suggestion. Data on the intensity of infection for most prevalent helminths is included in the revised MS. We also examined association of intensity of infections with hemoglobin/anemia and nutritional status. Accordingly, the revised statements in the MS are depicted as

“The mean egg per gram of stool for the most prevalent helminths seen among the children were 3003.5 (range = 24, 21, 600) for A. lumbricoides, 264.30 (range = 24, 3768) for S. mansoni, and 2618.8 (range = 24, 23904) for hookworms. The majority of A. lumbricoides (80.0%), hookworms (64.8%) and S. mansoni (45.6%) cases had light intensity infections. Close to 15% of children infected with hookworms and S. mansoni had heavy intensity infections. None of the children were heavily infected with A. lumbricoides.” (Lines 199-205).

The odds of anemia among children infected with S. mansoni increased with an increase in the epg of the parasite (aOR= 1.001, 95% CI= 1.00002, 1.001).” (Lines 238, 239).

“However, the odds of stunting (aOR = 2.87, 95% CI = 1.34, 6.18) and undernutrition (aOR = 2.28, 95% CI = 1.19, 4.40) among individuals infected with S. mansoni were greater for those with heavy than for those with light intensity infection. The odds of underweight (aOR = 3.77, 95% CI = 1.05, 13.48) and undernutrition (aOR = 3.36, 95% CI = 1.05, 10.77) among individuals infected with hookworm were also greater for those with moderate than for those with light intensity infection (lines 252-256).”

  • Page 2, Lines 83-86: This should be rewritten to express the aim of the study. What was the aim of the study?

Response: We have rewritten the aim of the study in lines 86 to 88 of the revised manuscript as ‘The aim of this is study was to examine the prevalence and intensity of intestinal helminth infections and their association with anemia, undernutrition, and academic performance among school age children in Maksegnit town, northwestern Ethiopia’.

  • Tables 1, 2, 3 – ‘Number’ should be clarified: number of what?

Response: We clarified ‘number’ in the column titles as Number examined (Table 1), Number infected and/or uninfected (Table 2 &Table 3), and Number infected (Table 4 and Table 5)

  1. Page 10, Lines 321-324: This sentence explains why the link between the intensity of infection and the parameters examined was not analyzed, but examination of intensity should be included in the “aim of the study”, and the results for the intensity of infection of individual parasite species should be given in the “Results” section. The information that the intensity of infection was low or moderate, given in the “Discussion” section, is insufficient.

Response: Thanks. Data on the intensity of infection is provided in the results section of the revised MS. Association of intensity of infections with hemoglobin/anemia and nutritional status is also examined and stated in Lines 86-88. In connection, the text in the discussion that justifies why the relationship between intensity of infection and anemia and nutritional status was not examined is removed.

Reviewer 2 Report

The reviewed paper deals with an attempt to assess the relationship of helminth infection among children in Maksegnit town (northern Ethiopia) with anemia, undernutrition, and academic performance. In my opinion, the manuscript does not fit the Aims&Scope of the Microorganisms, since it deals with parasitic worms that are macroparasites, not microorganisms. Other MDPI journals are more suitable, such as Parasitologia, Epidemiologia or Children.

It should be noted that the article is interesting and is devoted to relevant theme. I am of an opinion that the article could be published after some corrections.

Firstly, helminthiasis can cause not only cestodes, trematodes and nematodes, but also acanthocephalans, which also belong to parasitic worms (line 30). Helminthiasis can cause even monogeneans, which are parasitic worms too, but belong to ectoparasites. This should be taken into account.

The purpose of the study is not specified (line 83). The last sentence in Introduction should be redone something like this: “The aim (or purpose) of this study is …”

Authors should check the numbers in the tables and in the text. So in total (line 178), boys and girls give 91.8%. The remaining 8.2% are uninfected? It should be said. And further in the Results.

The authors should make all tables clearer. Thus the total number of studied children in the table 1 (1205) and 6–10 + 11–16 age groups (597+599) does not match; as well as sum of gender groups. Where did the rest of the children go? Matches only grade groups here.

In table 2 (and 3–6) the total number of children is 1188, not 1205. Why?

And the total number of children (1188) studied and the sum of sex (1180) groups and grade (1180) also do not match.

In section 3.3 (lines 207, 208) – 532+649 (also in table 3 – “uninfected” and “any helminth”), but in table 3 total number – 1188, does not match too. Here match only males+females – 1188.

And so on in all tables and text it is necessary to make the data more understandable.

It is desirable to give the Latin names of helminths in full in the tables.

Are significant differences highlighted in bold in the tables? Then it should be indicated in the notes under the tables.

And the traditional remark for my peer-reviewed articles for MDPI journals. In scientific papers, according ICZN (International Code of Zoological Nomenclature), at the first mention of an animal (helminth) species, its full Latin name with the author and year of description should be given (lines 39–41, 174–176 and others). Further, at the second mention, the Latin generic name of helminth is reduced to one letter, the author and the year are not given.

Latin names of helminths should be written in italics throughout the text (Lines 39, 174–176, 179, 182, 185, 186, 192 and further along the text).

Authors must different using of a dash “–“ and hyphen “-“: Dash without spaces means the interval “from–to” in numeric and character values: example: references [1–3], 576–740 million, 5–100%, 6–16 years etc.

or in case Kato–Katz technique.

Dash with spaces is used to denote dash in a text, period of the year, etc.: examples: ... where r – radius, m; ... in May – July.... Hyphen without spaces: socially-marginalized, meta-analysis, etc.

Spaces are needed in following cases: a) between number and the dimension. Example: 0.1 cm or 0.1 kg); b) between numbers and the multiplication sign or signs of “=”, “<”, “>”, etc. Example: n = 41, p = 0.005, WAZ < –2 and/or BAZ < –2; p < 0.0001, etc.

minus sign “–”, not “-”.

Lines 91, 182 – unnecessary spaces.

Line 272 – missing a space.

In my opinion, the manuscript can be published, but minor corrections are needed. The text of the manuscript is well written and does not require changes. Only numerical data require verification and explanation. Maybe make changes to the section Materials and Methods.

Author Response

Reviewer 2

The reviewed paper deals with an attempt to assess the relationship of helminth infection among children in Maksegnit town (northern Ethiopia) with anemia, undernutrition, and academic performance. In my opinion, the manuscript does not fit the Aims&Scope of the Microorganisms, since it deals with parasitic worms that are macroparasites, not microorganisms. Other MDPI journals are more suitable, such as Parasitologia, Epidemiologia or Children.

It should be noted that the article is interesting and is devoted to relevant theme. I am of an opinion that the article could be published after some corrections.

  1. Firstly, helminthiasis can cause not only cestodes, trematodes and nematodes, but also acanthocephalans, which also belong to parasitic worms (line 30). Helminthiasis can cause even monogeneans, which are parasitic worms too, but belong to ectoparasites. This should be taken into account.

Response: Thank you for the comments and suggested corrections. We included acanthocephalans in the list of parasites that cause Helminthiasis but opted not to include monogeneans to avoid duplication as they belong to the Platyhelminths/flat worms (line 30).

  1. The purpose of the study is not specified (line 83). The last sentence in Introduction should be redone something like this: “The aim (or purpose) of this study is …”

Response: The aim of the study is presented as “the aim of this  study was to examine prevalence and intensity of intestinal helminth infections and their association with anemia, undernutrition, and academic performance among school age children in Maksegnit town, northwestern Ethiopia.”in the revised MS (lines 86 to 89).

  1. Authors should check the numbers in the tables and in the text. So, in total (line 178), boys and girls give 91.8%. The remaining 8.2% are uninfected? It should be said.And further in the Results.

Response: The total prevalence of helminth infection among the study participants {in males (50.8%) and females (41.0%)} was 45.4%.  And the 54.6% were uninfected.

  1. The authors should make all tables clearer. Thus, the total number of studied children in the table 1 (1205) and 6–10 + 11–16 age groups (597+599) does not match, as well as sum of gender groups. Where did the rest of the children go? Matches only grade groups here.

Response: Many thanks for picking this. We have corrected the discordant in the number of children in each population group (age, gender, nutrition and anemia group) in Tables 1 to 3 in the revised MS.

  1. In table 2(and 3–6) the total number of children is 1188, not 1205. Why?

Response: Hemoglobin level was assessed for 1188 participants and the data/analysis reported in Table 2 and Table 3 was based on these individuals. Total number of participants is corrected in Tables 4, 5 and 6 as 1,205.

  1. And the total number of children (1188) studied, and the sum of sex (1180) groups and grade (1180) also do not match.

Response: This comment is appreciated. The discordant in the numbers of male and female, and age groups is corrected to match the total number (n=1188) in Tables 2 and 3 of the revised MS.

  1. In section 3.3 (lines 207, 208) – 532+649 (also in table 3 – “uninfected” and “any helminth”), but in table 3 total number – 1188, does not match too. Here match only males+females – 1188.

And so on in all tables and text it is necessary to make the data more understandable.

Response: Thank you. Discordants in the numbers are corrected in the revised manuscript (see the response above).

  1. It is desirable to give the Latin names of helminths in full in the tables.

Response: It is a valid comment. However, the full scientific names would increase the number of lines in headings making the tables hard to read/understand.

  1. Are significant differences highlighted in bold in the tables? Then it should be indicated in the notes under the tables.

Response: A footnote is added under each table that explains the bold results. It reads ‘bold values indicate that the differences are statistically significant”

  1. And the traditional remark for my peer-reviewed articles for MDPI journals. In scientific papers, according ICZN (International Code of Zoological Nomenclature), at the first mention of an animal (helminth) species, its full Latin name with the author and year of description should be given (lines 39–41, 174–176 and others). Further, at the second mention, the Latin generic name of helminth is reduced to one letter, the author and the year are not given.

Response: Thank you for the suggestion. As we are not sure about the formatting for the scientific names in the first mention in the microorganism journal, we would like to leave the decision to the editors.

  1. Latin names of helminths should be written in italics throughout the text (Lines 39, 174–176, 179, 182, 185, 186, 192 and further along the text).

Response: The scientific names are italicized throughout the revised manuscript as rightly suggested.

  1. Authors must different using of a dash “–“ and hyphen “-“: Dash without spaces means the interval “from–to” in numeric and character values: example: references [1–3], 576–740 million, 5–100%, 6–16 years etc. or in case Kato–Katz technique.

Response: Thanks for the suggestion. The ‘hyphen” is replaced with “dash without space” throughout the manuscript where necessary.

  1. Dash with spaces is used to denote dash in a text, period of the year, etc.: examples: ... where r – radius, m; ... in May – July.... Hyphen without spaces: socially-marginalized, meta-analysis, etc.

Response: The comment is well taken. The manuscript is checked for the correct use of dash with space and hyphen without space.

  1. Spaces are needed in following cases: a) between number and the dimension. Example: 0.1 cm or 0.1 kg); b) between numbers and the multiplication sign or signs of “=”, “<”, “>”, etc. Example: n = 41, p = 0.005, WAZ < –2 and/or BAZ < –2; p < 0.0001, etc.

Response: Spaces are made as appropriate throughout the manuscript.

  1. minus sign “–”, not “-”.

Response: we have replaced hyphen (-) with dash (–) where appropriate

  1. Lines 91, 182 – unnecessary spaces.

Response: unnecessary spaces removed in lines 91 and 182

  1. Line 272 – missing a space.

Response: Space added between the genus and species name in ‘E. vermicularis” and the word ‘range’ and the number ‘54.9 to 58.3%’.

  1. In my opinion, the manuscript can be published, but minor corrections are needed. The text of the manuscript is well written and does not require changes. Only numerical data require verification and explanation. Maybe make changes to the section Materials and Methods.

Response: Thank you for the constructive comments and your positive recommendation. We  revised the manuscript following your suggestions and hope you will find the corrections acceptable.

Reviewer 3 Report

In the manuscript submitted for review, examined the prevalence of intestinal helminth infections and their association with anemia, undernutrition, and academic performance among school age children in Maksegnit town, northern Ethiopia.

My comment:

The research results presented in the manuscript are a type of epidemiological research. It would be good if they were correlated with more laboratory tests, e.g. morphology, iron concentration, eosinophil count, etc. The work is well written, but it doesn't bear anything original. The strength of this article is the size of the study group.

I have doubts about the diagnosis of T. saginata and S. mansoni:

T. saginata
In the case of Taenia eggs, they are morphologically very similar and difficult to distinguish. Realizes that pork is rarely eaten in Ethiopia for religious and cultural reasons, the likelihood of T. solium is lower, but whether the eggs found in coproscopic examinations are definitely T. saginata. Were tapeworm proglots observed, which would allow to clearly distinguish these species?

S. mansoni

While the standard method of diagnosis for S. mansoni is microscopic identification of eggs in the faeces, eggs may be passed on sporadically or in small amounts. Should the diagnosis of this parasite not be supplemented with other methods, such as circulating cathodic antigen (CCA)?

From a diagnostic point of view, in this manuscript, I miss the application and comparison of other coproscopic methods.

It would be good if the authors of this article presented the most interesting results in graphic form.

The conclusion is very general. I think that in conclusion the authors should write specific information that results from their research.

Author Response

Reviewer 3

In the manuscript submitted for review, examined the prevalence of intestinal helminth infections and their association with anemia, undernutrition, and academic performance among school age children in Maksegnit town, northern Ethiopia. The research results presented in the manuscript are a type of epidemiological research. It would be good if they were correlated with more laboratory tests, e.g. morphology, iron concentration, eosinophil count, etc. The work is well written, but it doesn't bear anything original. The strength of this article is the size of the study group.

  1. I have doubts about the diagnosis of T. saginata and S. mansoni:
  2. saginata: in the case of Taenia eggs, they are morphologically very similar and difficult to distinguish. Realizes that pork is rarely eaten in Ethiopia for religious and cultural reasons, the likelihood of T. solium is lower, but whether the eggs found in coproscopic examinations are definitely T.saginata. Were tapeworm proglots observed, which would allow to clearly distinguish these species?

Response: Thank you for the valid comment. We replaced Taenia saginata with Taenia species to minimize potential identification errors.

  1. S. mansoni

While the standard method of diagnosis for S. mansoni is microscopic identification of eggs in the faeces, eggs may be passed on sporadically or in small amounts. Should the diagnosis of this parasite not be supplemented with other methods, such as circulating cathodic antigen (CCA)?

Response: Although we have used the standard method (Kato-Katz) applying two slides for each individual to improve the sensitivity of the microscope for the diagnose S. mansoni infection, we acknowledge that applying additional methods including CCA for the diagnosis of S. mansoni could improve the sensitivity of the test results. So, we have discussed this as a limitation of the study. The added text in the discussion reads ‘Furthermore, although we have used the standard method (two Kato-Katz slides) for the diagnosis ofS. mansoni infection, supplementing the test results with other techniques such as circulating cathodic antigen (CCA) could have improved the accuracy of the test results.”(line 356-359).

  1. From a diagnostic point of view, in this manuscript, I miss the application and comparison of other coproscopic methods.

Response: The main purpose of this study was to examine association of intestinal helminth infection with undernutrition, hemoglobin level and academic performance. Hence, we didn’t compare the performance of the techniques in diagnosing helminth infections.

  1. It would be good if the authors of this article presented the most interesting results in graphic form.

Response: Figures could be more interesting in reporting data. However, we found tables more appropriate to present the extended findings compared to figures.

  1. The conclusion is very general.I think that in conclusion the authors should write specific information that results from their research.

Response: We believe that the main findings of this study: (i) moderate prevalence of intestinal helminth infection in the study population; (ii) correlation of intestinal helminth infection with undernutrition, anemia and low academic performance describe the research outcomes adequately.

“In conclusion, there is a moderate prevalence of intestinal helminth infection among school age children in Maksegnit town, northern Ethiopia. This could increase the risk of anemia, undernutrition, and poor academic performance in the region.”

Reviewer 4 Report

This case report is well-written and provides details of the patients and disease prognosis. I have the following comments about the article.

  • In the result section, scientific names need to be italicized. Please see attached file.
  • Please rewrite line 16 in the abstract.
  • Words are repeated in line 141.
  • Correction in line 186 and other lines, please see attached file.
  • Minor typographical errors are there. Please see attached file. 
  • Table 1: Try to adjust table 1 and all heading should be in one line. In the present form, it is difficult to read the column heading and then correlate it with next page values. The same applies to other tables.
  • It would be great to provide a figure of the parasite slide to show the parasite in the samples.
  • Single infection with Helimenthis causes a higher decrease in hemoglobin in compared to double parasitic infection. Any explanation for that observation?

Author Response

Reviewer 4

This case report is well-written and provides details of the patients and disease prognosis. I have the following comments about the article.

  1. In the result section, scientific names need to be italicized. Please see attached file.

Response: Thanks for the right comment. Scientific names are italicized in the revised manuscript.

  1. Please rewrite line 16 in the abstract.

Response: The sentence in line 16, in the abstract, is  revised as “Hemoglobin level was measured using HemoCue machine”.

  1. Words are repeated in line 141.

Response: Thanks. The repeated word ‘ethics’ is deleted from the sentence in line 157.

  1. Correction in line 186 and other lines, please see attached file.

Response: We deleted the repeated word article ‘the’ from the sentence in line `201’ and added space between the genus and species name in ‘H.nana’

  1. Minor typographical errors are there. Please see attached file. 

Response: Thank you. The suggested corrections are made and  the manuscript is checked for any additional typographical error.

  1. Table 1: Try to adjust table 1 and all heading should be in one line. In the present form, it is difficult to read the column heading and then correlate it with next page values. The same applies to other tables.

Response: The headings in the original submission are formatted bythe editors. So, we would like to leave the decision to the editors.

  1. It would be great to provide a figure of the parasite slide to show the parasite in the samples.

Response: That would be great. Unfortunately, we didn’t take pictures of the parasites during examination as they were the common ones.

  1. Single infection with Helminths causes a higher decrease in hemoglobin in compared to double parasitic infection. Any explanation for that observation?

Response: Perhaps this could be because most infections were low intensity and the most common double infections were between A. lumbricoides and hookworm or S. mansoni. On the other hand, A. lumbricoides has little  impact on hemoglobin level of individuals.

Reviewer 5 Report

The paper reports data about helminthiases in school children from Ethiopia, related to some pathophysiological findings.  

The Authors have also found some cestoda (Taenia saginata, Hymenolepis nana), they should speak about these helminthic species in the introduction section 

Line 39 - Ascaris lumbricoides in italics

line 54 - Hookworms cause; everywhere change hookworm in hookworms

line 96 - anthelmintic; epidemiology

line 120 - please, modify presence in diagnostic stages

lines 173-176 and in following lines - Parasites proper names in italics; Enterobius vermicularis, Trichuris trichiura (also in Tables)

line 276 -anthelmintic drugs

Author Response

Reviewer 5

The paper reports data about helminthiases in school children from Ethiopia, related to some pathophysiological findings.

  1. The Authors have also found some cestoda (Taenia saginata, Hymenolepis nana), they should speak about these helminthic species in the introduction section 

Response: As for the STHs and S. mansoni, we have added data on the epidemiology of Taenia saginata, Hymenolepis nana in the introduction. The added text reads, “An estimated 95 to 135 million people in the world are also infected with Taenia species or Hymenolepis nana” [7,8].

  1. Magill AJ, Hill DR, Solomon T, Ryan ET. Hunter's Tropical Medicine and Emerging Infectious Disease. 2012 ISBN 978-1-4160-4390-4. DOI https://doi.org/10.1016/C2009-0-51934-4.
  2. Held M, Cappello M. Cestodes. Editor(s): Johnson LR, Encyclopedia of Gastroenterology, Elsevier, 2004, Pages 289-293, ISBN 9780123868602. https://doi.org/10.1016/B0-12-386860-2/00684-5.
  3. Line 39 - Ascaris lumbricoides in italics

Response: Thanks, We have italicized all scientific names including Ascaris lumbricoides in line 39, throughout the manuscript.

  1. Line 54 - Hookworms cause; everywhere change hookworm in hookworms

Response: We have replaced ‘hookworm’ with ‘hookworms’ throughout the manuscript

  1. line 96 - anthelmintic; epidemiology

Response: We have replaced the word ‘anihelminthic’ with ‘anthelminthic’in line 99. We also removed the repeated term article ‘the’ before the word Epidemiology in line 99.

  1. line 120 - please, modify presence in diagnostic stages

Response: We have rephrased the sentence in line 122/123 as ‘The slides were examined to detect the presence in diagnostic stage of intestinal helminth infections…”

  1. lines 173-176 and in following lines - Parasites proper names in italics; Enterobius vermicularis, Trichuris trichiura (also in Tables)

Response: We have italicized the scientific names throughout the manuscript.

  1. line 276 -anthelmintic drugs

Response: We have replaced anthelminthic with anthelminthic in line 280

Round 2

Reviewer 3 Report

Accept.

Reviewer 5 Report

The quality of the manuscript was improved, however some modifications in proper names of the parasites should be made

line 44 - Taenia, not Taenias

lines 134 and 192 - trichiura

everywhere in the manuscript - Taenia spp instead of Taenia species

Author Response

Dear Prof. Isabel Mauricio and Debra Liu,

We would like to thank you for handling the submission and review process. We also thank the reviewer for their careful review and constructive comments, which have helped to improve the manuscript further. We appreciate the fast response. We have made changes to the manuscript based on reviewer’s suggestions and describe these changes in the below paragraphs. We hope that you will find our responses acceptable, and we look forward to your decision.

Reviewer 5

The quality of the manuscript was improved, however some modifications in proper names of the parasites should be made

  1. line 44 - Taenia, not Taenias

Response: We have removed the letter ‘s’ at the end of the word Taenias in line 44.

  1. lines 134 and 192 – trichiura

Response: We have replaced “trichuria” with “trichiura” in line 134 and 192.

  1. everywhere in the manuscript - Taenia spp instead of Taenia species

Response: We have replaced ‘Taenia species’ with “Taenia spp’ throughout the manuscript.